# Survey of Viral Reactivations in Elite Athletes: A Case-Control Study

**DOI:** 10.3390/pathogens10060666

**Published:** 2021-05-28

**Authors:** Lari Pyöriä, Maarit Valtonen, Raakel Luoto, Wilma Grönroos, Matti Waris, Olli J. Heinonen, Olli Ruuskanen, Maria F. Perdomo

**Affiliations:** 1Department of Virology, University of Helsinki, 00290 Helsinki, Finland; lari.pyoria@helsinki.fi; 2Research Institute for Olympics Sports, 40700 Jyväskylä, Finland; maarit.valtonen@kihu.fi; 3Department of Pediatrics and Adolescent Medicine, Turku University Hospital and University of Turku, 20521 Turku, Finland; rajopi@utu.fi (R.L.); olli.ruuskanen@tyks.fi (O.R.); 4Paavo Nurmi Centre and Unit of Health and Physical Activity, University of Turku, 20520 Turku, Finland; wgegro@utu.fi (W.G.); olli.heinonen@utu.fi (O.J.H.); 5Institute of Biomedicine, University of Turku and Department of Clinical Microbiology, Turku University Hospital, 20520 Turku, Finland; mwaris@utu.fi

**Keywords:** elite athletes, viral immunity, TTV, HHV, immunocompetence

## Abstract

Exercise-induced immune perturbations have been proposed to increase susceptibility to viral infections. We investigated the replication of persisting viruses as indicators of immune function in elite cross-country skiers after ten months of sustained high-performance exercise. The viruses evaluated, nine human herpesviruses (HHVs) and torque teno virus (TTV), are typically restrained in health but replicate actively in immunosuppressed individuals. We collected sera from 27 Finnish elite cross-country skiers at the end of the competition’s season and 27 matched controls who perform moderate exercise. We quantified all the HHVs and—TTV via highly sensitive qPCRs. To verify equal past exposures between the groups, we assessed the IgG antibody prevalences toward HHV-4 (Epstein–Barr virus, EBV) and HHV-5 (human cytomegalovirus, HCMV). We found equal TTV DNA prevalences in athletes (63%) and controls (63%) and loads with respective geometric means of 1.7 × 10^3^ and 1.2 × 10^3^ copies/mL of serum. Overall, the copy numbers were low and consistent with those of healthy individuals. Neither of the groups presented with herpesvirus viremia despite similar past exposures to HHVs (seroprevalences of EBV 70% vs. 78% and HCMV 52% vs. 44% in athletes and controls, respectively). We found no evidence of increased replication of persistent viruses in elite athletes, arguing against impaired viral immunity due to high-performance exercise.

## 1. Introduction

Transient and cumulative perturbations in immune function following strenuous exercise have been proposed as the underlying cause for an increased risk of infection in elite athletes [1,2]. Indeed, higher frequencies of respiratory symptoms have been reported during and after competitions, although only a handful of studies have provided laboratory confirmation supporting the clinical assessment [3,4,5,6,7]. We have previously observed up to a sevenfold increase in the relative risk to viral respiratory infections in elite cross-country skiers compared to healthy controls [7,8], with long-haul travel, social housing, and mass-gatherings as important risk factors.

To circumvent confounding epidemiological variables, in the present study we examined reactivations of pre-acquired viral infections. The aim was to investigate whether sustained strenuous exercise would affect immune function and result in the replication of persistent viruses.

Many viruses that infect us during childhood remain latent in the tissues, their replication being repressed through continuous surveillance by both innate and adaptive arms of the immune system [9,10,11]. This is the case for the nine human herpesviruses (HHVs) and torque teno virus (TTV)that can reactivate upon immunosuppression or specific environmental triggers. HHV reactivations can be asymptomatic or present with clinical manifestations ranging from local (e.g., HSV-1&2 and VZV mucocutaneous lesions) to life-threatening conditions (e.g., HCMV disease or EBV post-transplantation lymphoproliferative disorder). While local bursts of replication may be common even in healthy individuals, the presence of HHV DNA in sera is a strong indicator of immunosuppression [12,13,14,15,16,17]. TTV infections, on the other hand, result in asymptomatic, lifelong low-level viremia that can be significantly increased in immunosuppression [18,19] and immunosenescence [20,21].

We evaluated the genomic frequencies and copy numbers of these ten viruses in the sera of elite cross-country skiers from the national ski team of Finland, participating in the 2019 National Championships in Äänekoski, Finland, an event held at the end of the skiers’ competition season (5 months). As controls, we included age and gender-matched individuals who perform moderate exercise. Moreover, we determined the serological status toward HHV-4 (Epstein–Barr virus, EBV) and HHV-5 (human cytomegalovirus, HCMV) to verify equal past exposures to these viruses in the groups.

## 2. Results

### 2.1. Exercise Load

The participants were selected based on similar high training loads. The mean yearly training volume of the 26 elite cross-country skiers was 766 h (range 580–902), i.e., on average 15 h per week. The training consisted typically of 90% endurance (low, moderate, or high intensity), 8% strength, and 2% speed. The modes of training included running, cycling, and skiing/roller skiing. During the 5-month competition season, the athletes participated in 30–60 (median 35) events. In the matched control group, the exercise load was less than 6 h per week.

### 2.2. Detection of Viral DNAs in Sera

We quantified TTV loads in sera via a highly sensitive qPCR that recognizes all known genogroups of TT viruses.

We detected TTV DNA in 63% of elite athletes (17/27) and 63% of controls (17/27) (95% CI ranges 44–78%). The viral loads, calculated from individuals with TTV viremia, were on average 1.7 × 10^3^ copies/mL of serum in athletes and 1.2 × 10^3^ copies/mL of serum in the control group (geometric mean, 95% CI ranges 9.0 × 10^2^–3.1 × 10^3^ and 5.6 × 10^2^–2.6 × 10^3^ copies/mL of serum, respectively). The differences in the viral DNA prevalences or loads lacked statistical significance (*p* = 1.00 and *p* = 0.519, respectively; Figure 1).

We detected none of the HHV DNAs in the sera of either the elite athletes or the controls, ruling against reactivation.

### 2.3. Past Immunity against Herpesviruses

Given the ubiquitous nature of the viruses studied and the homogeneous demographics of the target and control groups, similar past exposures were to be expected. Nevertheless, to verify this, we examined the IgG antibody prevalences toward EBV and HCMV.

We found similar seroprevalences in the elite athletes and controls (Figure 2), being respectively the EBV IgG positivity 70% (19/27) and 78% (21/27), (95% CI ranges of 52–84%, and 59–89%, respectively), and the HCMV IgG positivity 52% (14/27) and 44% (12/27), (95% CI ranges of 34–69% and 28–63%, respectively; Figure 2).

There were no statistically significant differences in the serological findings (*p* = 0.757 for EBV and *p* = 0.786 for HCMV).

## 3. Discussion

Recurrent flares of immunosuppression following strenuous exercise may cumulatively increase the susceptibility to infection in elite athletes [2]. We investigated latent viruses as integral markers of immune function since all components of the immune system are required to contain reactivation [10,22].

We examined the viral loads of TTV and the HHVs in the sera of elite cross-country skiers after 10 months of heavy training and competitions, following probably the highest training volume of all athletes [23]. The aim was to evaluate whether sustained high-performance exercise could compromise the control of these otherwise normally suppressed viruses [9].

TTV infections are acquired soon after birth and result in chronic asymptomatic viremia [24] that can be significantly elevated in immunosuppression [18,25] and immune senescence [20,21,26]. Indeed, in immunocompromised individuals, such as transplant recipients [17], the average copy numbers can be as high as 10^10^/mL of plasma [18]. For this reason, the kinetics of this virus in the blood continue to be extensively evaluated as a diagnostic and prognostic marker of immune function in different conditions [21,27,28,29,30,31,32], and in the monitoring of immunosuppressive treatments [17,19,22,33].

In the present study, the TTV DNA prevalences found were similar in both elite athletes and control subjects, and the viral loads compatible with those of healthy individuals [25,26].

Herpesviruses establish latency with occasional cycles of active replication being triggered, among others, by physical and emotional stress. In athletes, EBV DNA has been previously investigated in peripheral blood leukocytes [34] and saliva [35,36,37]. However, these sample types are more likely to reflect latency or intermittent stages of replication [38,39,40,41], and positivity is common in the general population. Gleeson et al. [35], for example, found a correlation between EBV DNA shedding in the saliva of elite swimmers and upper respiratory symptoms; however, low-level replication of EBV also occurs in the saliva of healthy individuals [40] and this was not accounted for in the study by inclusion of a control group.

In contrast, the finding of HHV DNAs in cell-free fluids, such as serum, is uncommon in healthy individuals and consequently is a specific indicator of immune deficiency [12,42,43].

We found no cases of HHV DNA in the sera of either the elite athletes or controls. In blood cells, Hoffmann et al. [34] detected statistically significantly higher loads of EBV DNA in athletes compared to healthy controls, albeit in both groups the copy numbers were low. The differences reported may mirror ongoing cellular processes, favoring low-level replication that may not be reflected in sera, if not resulting in lysis of the host cell. Thus, follow-up studies evaluating sequential viral loads in both the cell-free and cellular fractions of blood in larger cohorts are warranted [44].

Given the high seroprevalences of most of the viruses studied here [9,45,46] and the analogous demographics of the target and control groups, it is feasible to assume corresponding past exposures to these viral infections. Nevertheless, we verified similar IgG seroprevalences toward EBV and HCMV, which were in line with those of the general population of similar ages (78–95% for EBV and 34–72% for HCMV in Western countries) and other athletes (78% and 36% for EBV and HCMV, respectively) [34,45,47,48,49,50].

In the present study, we investigated the reactivation of common pre-acquired viral infections at the end of the competition’s season, thus accounting for the long-term sequelae on immune competence. Taken together, we found no evidence of viral reactivation in connection to sustained high-performance exercise.

## 4. Materials and Methods

The study included 27 athletes belonging to the Finnish cross-country ski team participating in the 2019 National Championships in Äänekoski, Finland. The participants were selected based on their similar training loads. The training season of the athletes started on 1 May and the competition season on 1 November, accounting for 10 months of heavy physical stress.

The training load of the athletes was collected from the day-to-day training diary data of the previous 10 months. The information of one athlete could not be obtained. For every other athlete, one healthy, moderately physically active (<6 h/week) control subject was recruited among the students and staff of Turku University Hospital and Turku University. The controls (*n* = 27) were matched for age (±2 years), gender, and the number of children younger than 5 years of age in the household (Table 1). The serum samples and health-related information were collected by a study nurse from the athletes before the competition (28 March 28 and 1 April 2019), and from the controls between 2 and 11 April 2019. The serum samples were immediately frozen upon collection.

The DNA was extracted from 200 µL of serum using the QIAamp DNA Blood Mini Kit (Qiagen) following the manufacturer protocol. The final elution volume was in 80 µL AE buffer.

The nine HHVs (human simplex viruses 1 and 2, varicella-zoster virus, EBV, HCMV, human herpesviruses 6A-B and 7, Kaposi’s sarcoma-associated herpesvirus) were analyzed via quantitative multiplex PCRs [51] and TTV DNA via qPCR as described [52].

The EBV and HCMV IgG analyses were performed with the human anti-Epstein–Barr viral capsid antigen (VCA) IgG ELISA Kit and anti-cytomegalovirus IgG ELISA kit (Abcam) according to the manufacturer protocols. The standard units were calculated as specified by the manufacturer.

Fisher’s exact test was used to compare the TTV DNA prevalences and the EBV and HCMV IgG seroprevalences between the groups. Wilson’s score interval was used to calculate the 95% confidence interval for binomial proportions. The viral loads from TTV DNA-positive individuals were log10-transformed for statistical analyses and compared via Student’s *t*-test. The analyses were done with SPSS (V.23) and *p* > 0.05 interpreted as not statistically significant. R-studio (v.1.2.5033) and Excel (v.2002) were used to create the figures.

## Figures and Tables

**Figure 1 pathogens-10-00666-f001:**
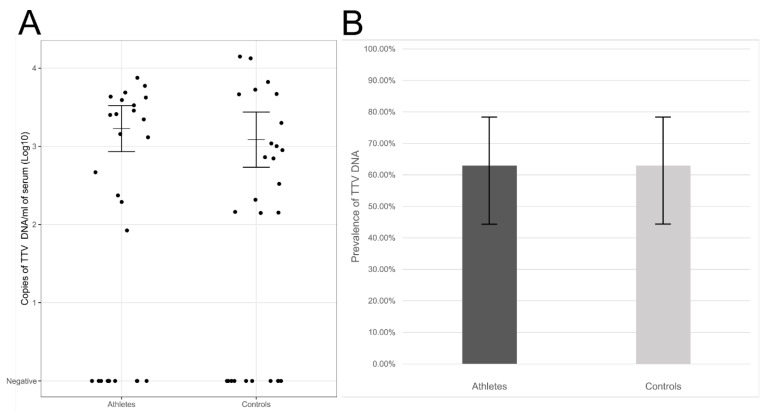
(**A**) TTV DNA viral loads in sera of elite athletes and controls. Represented are in the *y*-axis, the log10 copies of TTV DNA/mL of serum in athletes (left) and controls (right). Horizontal lines represent the geometric means and whiskers the 95% confidence interval of the viral loads calculated from the TTV DNA-positive individuals. (**B**) TTV DNA prevalences in sera of athletes (dark grey) or controls (light grey). The differences were not statistically significant (*p* > 0.05).

**Figure 2 pathogens-10-00666-f002:**
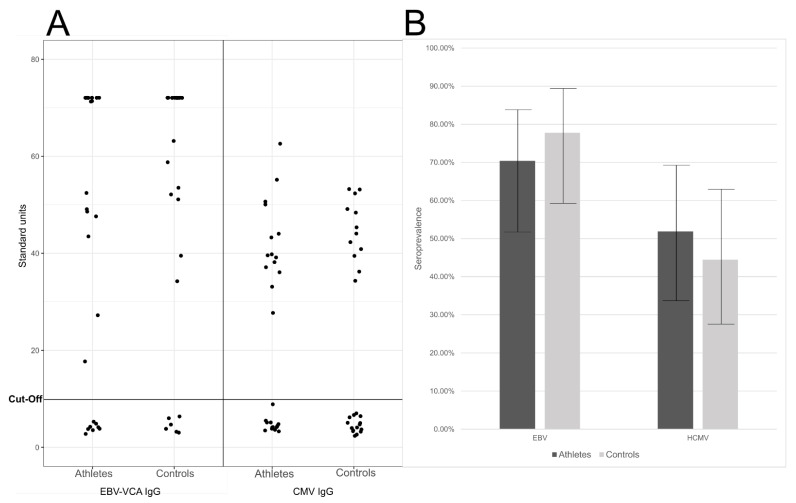
(**A**) EBV VCA (viral capsid antigen) IgG and HCMV IgG levels. Represented are the calculated standard units (*y*-axis) for each of theathletes (left) and controls (right). (**B**) Seroprevalences of EBV and HCMV. Represented are the percentage of IgG positive individuals (*y*-axis) in athletes (dark grey) and controls (light grey). The whiskers represent the 95% confidence interval of the prevalences. The differences were not statistically significant (*p* > 0.05).

**Table 1 pathogens-10-00666-t001:** Clinical characteristics of the study population.

	Athletes (*n* = 27)	Controls (*n* = 27)
Age	27.1 (20.0–40.9)	27.4 (20.7–40.4)
Female	13 (48%)	13 (48%)
BMI	22.05 (18.0–24.8)	24.0 (18.4–35.2)
Children (<5 years) at home	3 (11%)	3 (11%)
Training load, h/week (*n* = 26)	15 (11–17)	<6

Values represent means (ranges) or numbers (proportions %).

## Data Availability

All data generated or analyzed during this study are included in this article. Additional information will be available from the corresponding author upon reasonable request.

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
