# Peer review of "Survey of Viral Reactivations in Elite Athletes: A Case-Control Study"

_pathogens, 2021, doi:10.3390/pathogens10060666_

Round 1

Reviewer 1 Report

The manuscript present interesting data on DNA viruses reactivation upon heavy sport training. It is very interesting, from sports medicine point of vue.

I have minor concerns about the introduction. I think there should be more background on the viruses detected in the study and their possible reactivation. There must be some research on reactivation of those viruses in people.

Other comments:

line 122 genoprevalence  change to TTV- DNA prevalence

line 148 what are detailed numbers of seroprevalence of those viruses in Finnland and other countries?

EBV-VCA abreviation should be explained earlier (not only in method section)

Author Response

The manuscript present interesting data on DNA viruses reactivation upon heavy sport training. It is very interesting, from sports medicine point of vue.

I have minor concerns about the introduction. I think there should be more background on the viruses detected in the study and their possible reactivation. There must be some research on reactivation of those viruses in people.

We thank the Reviewer for his assessment of our manuscript and for this comment. We have now added more background in lines 56-64 related to the targeted viruses.

Other comments:

line 122 genoprevalence  change to TTV- DNA prevalence

This has now been changed, in the new version appears in line 131

line 148 what are detailed numbers of seroprevalence of those viruses in Finnland and other countries?

The seroprevalences in Western countries in similar age groups to our study cohort and seroprevalences from studies with athletes have now been added to the text lines 157-158.

EBV-VCA abreviation should be explained earlier (not only in method section).

The abbreviation has now been opened up in the methods (line 195) and also in Fig.2 legend, where it first appears in the text

Reviewer 2 Report

The study is of some interesting. And the below revisions are needed.

Major Revision

#1:

n=27 vs 27 is too small to allow for any definite conclusions. In contrast, 44 elite athletes and 68 controls were analysed in the authors’ previous study. Why the authors selected only 27 athletes for the study? Please explain in the manuscript.

Minor Revision

#2:

Line 47: Unpublished article (reference 8 seems “under review”) must not be referred.

#3:

Line 216: Please mention the details (institution and/or department) of “Professor Klaus Hedman”.

Author Response

The study is of some interesting. And the below revisions are needed.

Major Revision

#1:

n=27 vs 27 is too small to allow for any definite conclusions. In contrast, 44 elite athletes and 68 controls were analysed in the authors’ previous study. Why the authors selected only 27 athletes for the study? Please explain in the manuscript.

 We thank the Reviewer for his assessment of our manuscript and for this comment.

We agree that 27 is a rather small cohort size. However, the previous study mentioned by the reviewer was performed on Team Finland during the 2018 PyeongChang Olympic Games. The Team included 44 athletes from different sport disciplines who differ markedly in their training loads. The current study included only elite cross-country skiers, who have the highest training loads, making the study group homogenous in relation to their physical stress. This is critical for the evaluation of the immunological function in connection to long-term strenuous exercise.

Related to this, precisions have now been introduced in the text in lines 74, 165-166.

Minor Revision

#2:

Line 47: Unpublished article (reference 8 seems “under review”) must not be referred.

At the moment of submission of this manuscript, the work referred was under revision at PlosOne. Now the work has been accepted and is published as Valtonen M et al., (2021) Increased risk of respiratory viral infections in elite athletes: A controlled study. PLoS ONE 16(5): e0250907.

This reference has been now appropriately added to the manuscript file ( ref 8, line 47)

#3:

Line 216: Please mention the details (institution and/or department) of “Professor Klaus Hedman”.

The affiliation details of Prof. Klaus Hedman have now been added to the acknowledgments, line 227-228

Round 2

Reviewer 2 Report

Thanks for the revisions. The revisions are satisfactory.